# Observation of replica symmetry breaking in disordered nonlinear wave propagation

Davide Pierangeli[1], Andrea Tavani[1], Fabrizio Di Mei[1], Aharon J. Agranat[2], Claudio Conti [1,3] & Eugenio DelRe[1,3]

A landmark of statistical mechanics, spin-glass theory describes critical phenomena in disordered systems that range from condensed matter to biophysics and social dynamics. The most fascinating concept is the breaking of replica symmetry: identical copies of the randomly interacting system that manifest completely different dynamics. Replica symmetry breaking has been predicted in nonlinear wave propagation, including Bose-Einstein condensates and optics, but it has never been observed. Here, we report the experimental evidence of replica symmetry breaking in optical wave propagation, a phenomenon that emerges from the interplay of disorder and nonlinearity. When mode interaction dominates light dynamics in a disordered optical waveguide, different experimental realizations are found to have an anomalous overlap intensity distribution that signals a transition to an optical glassy phase. The findings demonstrate that nonlinear propagation can manifest features typical of spin-glasses and provide a novel platform for testing so-far unexplored fundamental physical theories for complex systems.

[1] Dipartimento di Fisica, Università di Roma "La Sapienza", 00185 Rome, Italy. [2] Applied Physics Department, Hebrew University of Jerusalem, 91904 Jerusalem, Israel. [3] Institute for Complex Systems, ISC-CNR, 00185 Rome, Italy. Correspondence and requests for materials should be addressed to D.P. (email: davide.pierangeli@roma1.infn.it)

D isorder in physical systems can introduce frustration among its interacting constituents. This implies that a large number of energetically equivalent and ergodically separated states are accessible in phase space. The condition in which these states dominate dynamics is generally known as the glassy state[1,2]. From the equilibrium perspective of the spin-glass theory, the transition to a glassy phase is signaled by replica symmetry breaking (RSB), that is, a change in the statistical distribution of the overlap between measurements in different realizations of the dynamics[3,4]. This Parisi overlap is the order parameter that indicates the transition to a RSB phase dominated by an energetic landscape. In spite of huge theoretical efforts, the replica breaking scenario has been only recently observed in photonics in the spectral features of random lasers[5–8] and multimodal laser emission[9]. In the latter case, although structural disorder is absent, frustration between modes emerges directly in their interaction. In random lasers, a spin-glass approach to the cavity modes of the electromagnetic field predicted how the competition of quenched disorder and nonlinearity induces glassy mode-locked regimes with many degenerate lasing states[10–15]. The resulting RSB phenomenology has also been found to be robust with respect to an averaging over different realizations of the disorder[16]. However, at variance with the case of nonlinear propagation, lasing dynamics require the influx of external energy[17]. The fact that a closed Hamiltonian system, and, in particular, a nonlinear wave propagation, can support RSB is still an open question with great relevance in fields such as nonlinear optics, polaritonics and Bose–Einstein condensates[13].

Generally, the dynamics of disordered nonlinear waves admits a non-equilibrium statistical mechanics description in terms of wave turbulence[18–20], which involves phenomena such as wave condensation[21–23] and strong turbulence of coherent and incoherent structures[24–34]. In particular, interacting localized modes propagating in nonlinear disordered media[35–37] and multiple laser filaments in gases[38,39] may lead to optical states whose complexity resembles glassy phases. This suggests that the statistical properties of coupled nonlinear optical waves can be investigated statically by means of Hamiltonian models with quenched disordered interactions. In this respect, fields propagating under a generalized nonlinear Schrödinger equation (GNLSE) with disorder should sustain a transition to a glassy behavior. According to theoretical predictions, replica symmetry breaking for equivalent realizations of the optical dynamics takes place for increasing nonlinearity, even for small degrees of disorder[13].

Here we report the observation of the breaking of replica symmetry in nonlinear optical propagation. Shot-to-shot fluctuations and spatial correlations of the optical field are investigated in a photorefractive disordered slab waveguide across the transition that leads from coherent to optically turbulent propagation, where strong variations of the speckle pattern and the degree of spatial correlations set in. In remarkable agreement with the general theories of spin-glasses, the Parisi overlap probability distribution function undergoes a radical change into a double-peaked distribution as the nonlinearity exceeds a threshold value. Replica symmetry breaking here indicates a global locking of several spatial modes so that completely anticorrelated states may emerge from equivalent conditions, the signature that different metastable states underlie dynamics.

## Results

**Nonlinear optical propagation in a disordered waveguide**. To investigate RSB in nonlinear propagation we make use of the large optical nonlinearity of disordered ferroelectric crystals in proximity of their structural phase transition[28,37,40,41].

Specifically, we exploit a disordered micrometric-sized photorefractive slab waveguide of potassium-lithium-tantalate-niobate (Fig. 1a, b). The experimental geometry of our setup is sketched in Fig. 1a and detailed in Methods. This system has been also recently used to experimentally demonstrate light beams undergoing antidiffraction and negative mass dynamics[42]. In our experiments, spatial inhomogeneities spontaneously arising in the slab constitute a weak linear disordered optical potential whose modes are mainly delocalized. The linear interaction between these optical modes is weak and nonlinearity is needed to couple them all. As shown in Fig. 1c and detailed in Methods, distinct realizations of the experiment present a different spatial distribution of disorder. Structural disorder changes from one replica ("shot") to another whereas it is fixed on the time scale of each single realization of the dynamics. Therefore, the replica symmetry breaking phenomenology we report hereafter has to be considered as averaged over different realizations of the disorder in analogy with results in non-static random laser systems[16]. The waveguide itself excludes dynamics along the transverse coordinates $y$, simplifying data analysis and allowing us to adopt a one-dimensional GNLSE model for wave propagation (see Supplementary Note 1). In our experiment, the degree of disorder is fixed by the experimental conditions so that the strength of the nonlinearity plays the role of an inverse temperature[13]. The strength of the nonlinearity is controlled by the time the waveguide is exposed to the input light beam (Supplementary Note 2). This relies on the nature of the photorefractive nonlinearity, that is noninstantaneous and accumulates in time as a photogenerated space-charge field builds up[43]. Observations at different times correspond to beam propagation for increasing nonlinearity, up to saturation. The macroscopic time scale of the process is fixed by the input power and the applied voltage[28].

**Transition to an optical glassy phase**. The spatial intensity distribution $I(x)$ detected as a function of the nonlinearity is reported in Fig. 2a, b for input powers $P = 0.2$ mW, $P = 5$ mW and applied voltage $V = 200$ V. We observe a phase transition to a disordered state that corresponds to the loss of spatial coherence. The transition is characterized by a large increase in the width of the spatial Fourier spectrum $I(k)$ (dots in Fig. 2c) and corresponding collapse of the long-range mean intensity autocorrelations (squares in Fig. 2c). The process presents strong shot-to-shot fluctuations, where each shot consists in repeating the experiment under analogous conditions (see Methods). Samples of single-shot spectra are reported in Fig. 2d above and below the transition threshold $t_c$. The averaged spectrum exhibits a change from a peaked to a broad distribution. The behavior of the spectrum in Fig. 2d presents a strong analogy with the static structure factor in amorphous materials and soft matter in the transition from an ordered to a glassy state. Interestingly, the transition occurs as fast as the input intensity: $P$ can thus be considered an equivalent of the quenching rate in structural glasses, where the transition temperature is known to decrease with supercooling[44].

**Evidence of replica symmetry breaking**. To demonstrate the emergence of replica symmetry breaking, we analyze the statistical properties of shot-to-shot correlations above and below the dynamic glass transition, at the threshold value of nonlinearity $t_c$ ($t_c \simeq 18$ s and $t_c \simeq 5$ s for $P = 0.2$ mW and $P = 0.5$ mW, respectively). Indeed, RSB is identified by a specific order parameter: the Parisi overlap $q$ that quantifies the correlation between fluctuations in different replicas (see Supplementary Note 1). As we measure the output intensity distribution, a natural choice to

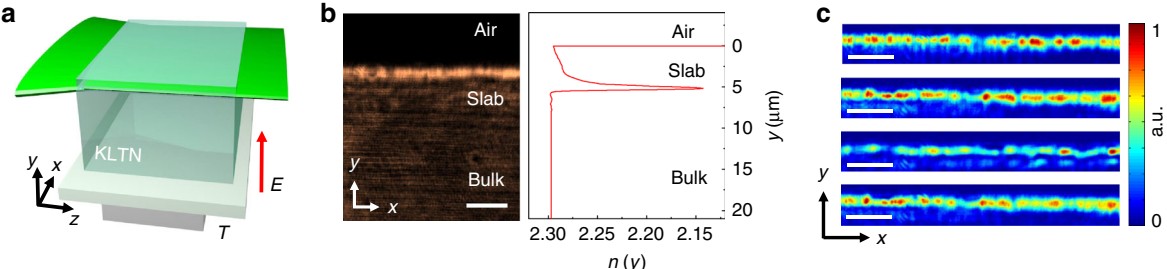

**Fig. 1** Experimental setup. **a** Scheme for optical wave propagation in a photorefractive slab waveguide embedded in a KLTN crystal. **b** Output image of the linear transmission through the sample and corresponding refractive index profile (see Methods). **c** Examples of the detected intensity in linear conditions for distinct realizations of the experiment showing different realizations of the disorder in the slab along x. Scale bars are 50 μm

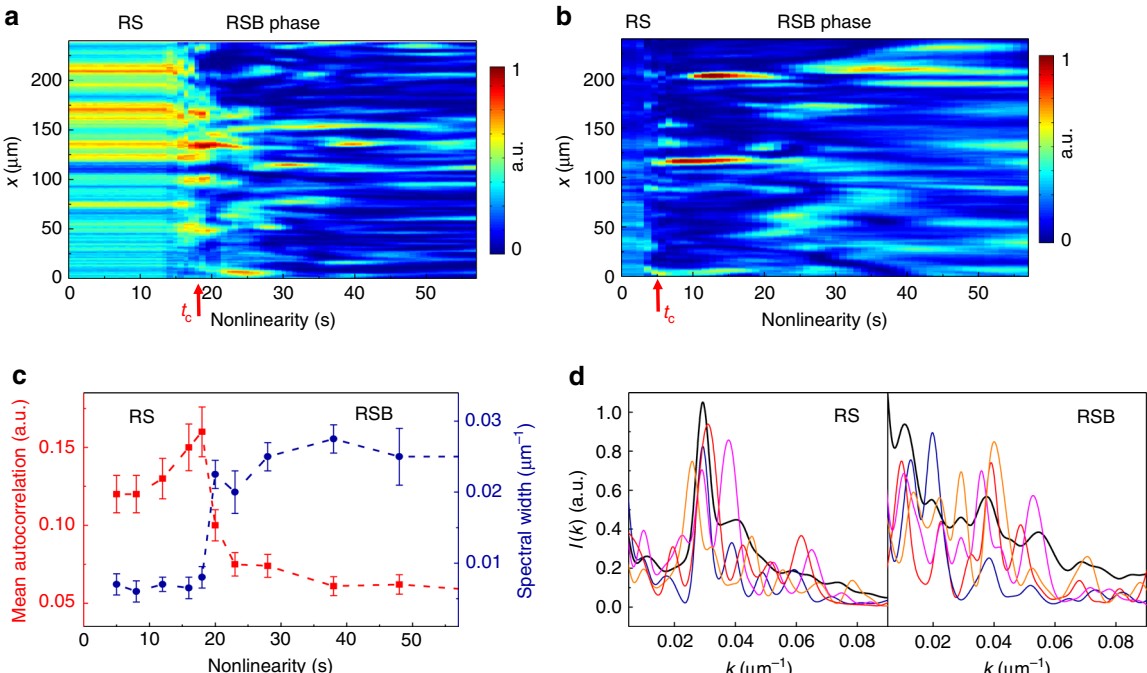

**Fig. 2** Loss of spatial coherence and shot-to-shot fluctuations. **a**, **b** Detected intensity distributions $I(x)$ at the slab output for increasing nonlinearity and input powers **a** $P = 0.2$ mW, **b** $P = 5$ mW. Both observations show a dynamic phase transition at $t_c$ (marked by a red arrow) to a spatially incoherent state. **c** Width of the Fourier spectrum and mean intensity autocorrelation as a function of the nonlinearity for the case in **a**. Dashed lines serve to guide across the transition. Error bars are given by the finite-size of the imaged area. **d** Sample of shot-to-shot flucutations of the spectral intensity: single-shot spectra (color lines) and averaged spectrum (black line) at $t \simeq 16$ s and at $t \simeq 40$ s. Replica symmetric (RS) and replica symmetry breaking (RSB) phases are indicated

characterize each realization is to consider the spatial auto-correlation function

$$g_\alpha(x) = \sum_{x'}^{R} I_\alpha(x') I_\alpha(x' + x), \qquad (1)$$

where $R$ is a cut-off length. Mode interaction and locking generally result in typical features of the autocorrelation function. In fact, possible phase correlations between different spatial points in the transverse field contribute as constructive and destructive interference effects at a distance $x$. Moreover, as this quantity reflects only global properties of the optical field, it is not affected by changes in the alignment of the optical setting from shot to shot that can alter the overlap evaluation (see Supplementary Note 1). We define the single-shot fluctuation as $\Delta_\alpha(x) = g_\alpha(x) - \overline{g}(x)$, where $\overline{g}(x)$ is the correlation averaged over all realizations. The experimentally accessible overlap

parameter $q_{\alpha\beta}$, that quantifies the similarity between shot-to-shot intensity fluctuations, is thus

$$q_{\alpha\beta} = \frac{\sum_x^R \Delta_\alpha(x) \Delta_\beta(x)}{\sqrt{\sum_x^R (\Delta_\alpha(x))^2} \sqrt{\sum_x^R (\Delta_\beta(x))^2}}. \qquad (2)$$

The measured intensity over $N = 120$ independent realizations of the dynamics is used to calculate the set of all $N(N-1)/2$ values of $q_{\alpha\beta}$, so as to determine their probability distribution $P(q)$ for different values of nonlinearity. The experimental order parameter $P(q)$ here defined is a coarse graining of the theoretical distribution of the overlap between mode amplitudes, a distribution that is not directly accessible and forms the fundamental

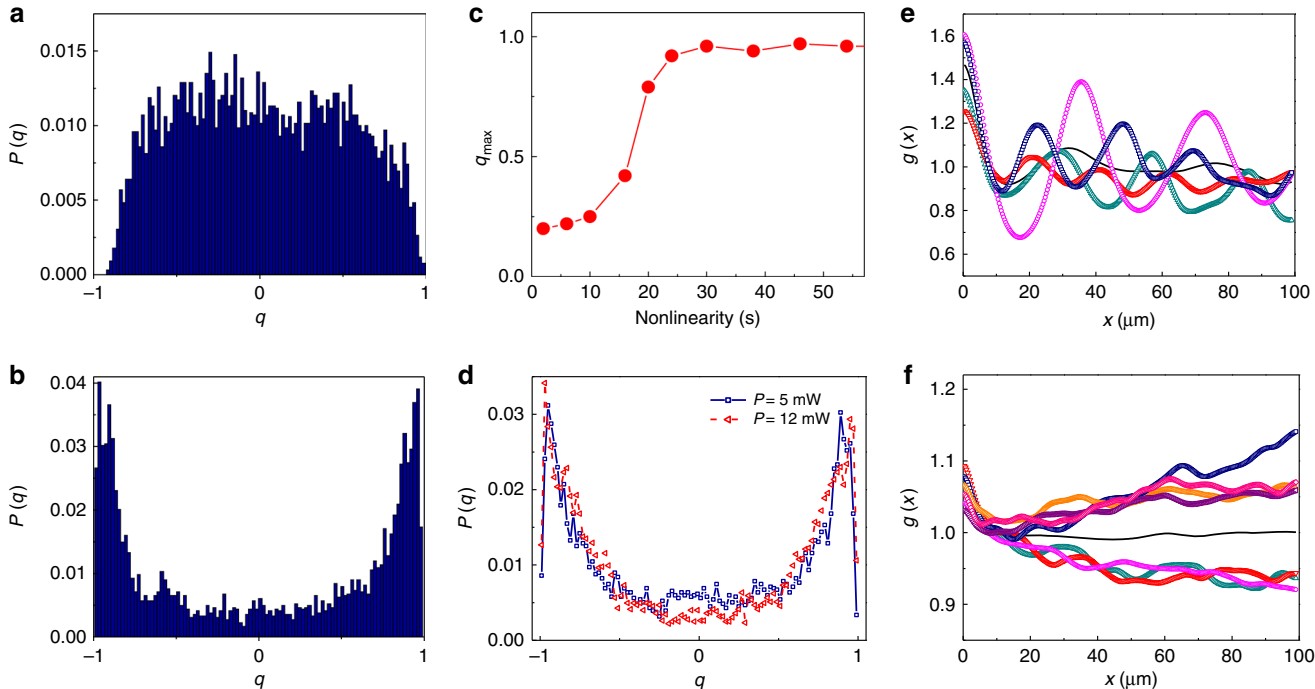

**Fig. 3** Evidence of replica symmetry breaking. **a** Overlap distribution measured for moderate nonlinearity $t \simeq 18$ s and **b** in the highly-nonlinear regime at $t \simeq 40$ s (input power $P = 0.2$ mW, cut-off scale $R = 100$ μm). **c** Overlap $q_{max}$, corresponding to the maximum in $P(|q|)$, as a function of the interaction strength. **d** Same as in **b** for data sets collected using different powers of the input wave ($P = 5$ mW, $P = 12$ mW). **e** Glass transition in the correlation functions shown by $g_\alpha(x)$ up to $R = 100$ μm for different replicas (color dots) at $t \simeq 18$ s and **f** at $t \simeq 40$ s. The two well-separated groups of states in **f** form the basis for replica symmetry breaking. The black solid line in **e**, **f** indicates the average correlation $\bar{g}(x)$

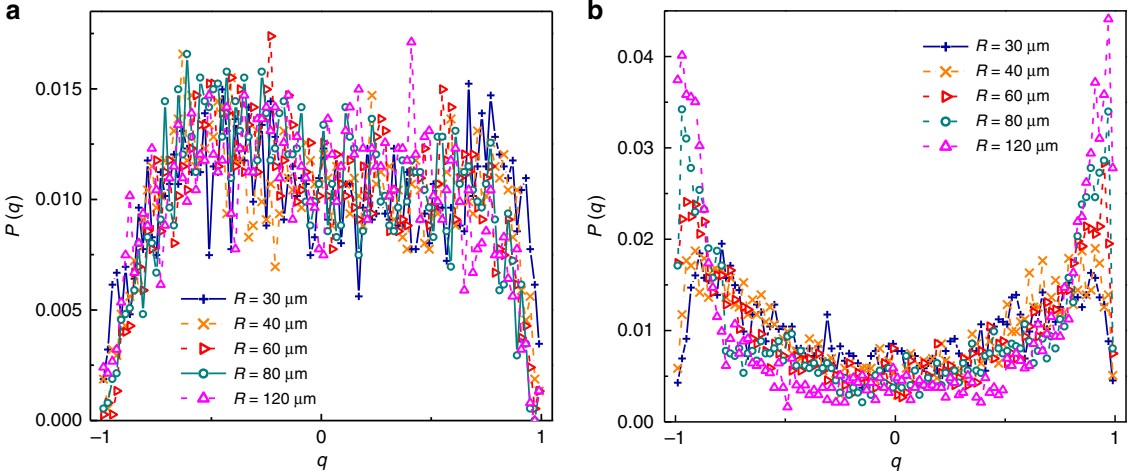

**Fig. 4** Stability of replica symmetry breaking. **a** $P(q)$ for different cut-off scales $R$ at $t < t_c$ (paramagnetic-like phase) and **b** at $t > t_c$ (spin-glass-like phase)

quantity that describes the glassy phase transition for light in terms of RSB (Supplementary Note 1).

Results are reported in Fig. 3. At moderate nonlinearity (Fig. 3a) the overlap distribution is centered around zero, which indicates that the correlation between field amplitudes in different points is an independent variable and modes do not interact strongly. The behavior drastically changes as modes are strongly coupled by the nonlinearity. As reported in Fig. 3b, a non-trivial overlap distribution emerges for $t > t_c$; the order parameter $q$ assumes all of its possible values ($P(q) > 0$) and the largest ones are particularly enhanced. $P(q)$ shows that, under the same experimental conditions, the shot-to-shot correlations are extremely sensible to the selected measurements. This is the

signature of the breaking of replica symmetry. It identifies an optical glassy phase in which the interplay between disorder and nonlinearity leads to locked intensity fluctuations. In Fig. 3c we show the maximum overlap $q_{max}$, that is, the absolute value for which we observe the maximum of $P(q)$, as a function of the nonlinearity. In agreement with the overall change in the overlap distribution, $q_{max}$ significantly grows around $t \simeq t_c$, indicating a phase transition that coincides with the one in Fig. 2b. Although the breaking of the replica symmetry cannot be rigorously characterized across this transition, the shape of the $P(q)$ in Fig. 3b suggests a one-step plus full replica symmetry breaking scenario (1RSB + FRSB) continuous in the order parameter $q$[9,15]. In fact, if a single-peaked distribution centered around $|q| \approx 1$

would indicate two distinct groups of states underlying the dynamics (1RSB), the region at $|q| \sim 0$ is not completely depleted, as compatible with FRSB (hierarchy of states). The observation of RSB does not depend on the input intensity; in Fig. 3d we report the $P(q)$ for $P = 5$ mW and $P = 12$ mW. The qualitative shape of the distribution remains unaltered.

A physical picture underlying RSB in the present case can be given considering the behavior of each autocorrelation function. In Fig. 3e, f we report different $g_\alpha(x)$ as examples of both the replica-symmetric and non-replica-symmetric case. The first presents oscillations with variable amplitude and phase (Fig. 3e); the corresponding fluctuations depend on the replica selected and vary with the relative distance $x$, so that their mutual correlation is randomly distributed. Differently, as shown in Fig. 3f, two distinct behaviors can be found in the glassy phase: at every distance a generic realization can be either more or less correlated than the average. These two trends can be thought of as two groups of states acting as separate dynamical attractors; in terms of spin variables, they can be thought of as ferromagnetic and anti-ferromagnetic dominated configurations. Each realization tends to fall into one of these two branches, so that fluctuations from shot to shot appear either completely correlated ($q \approx 1$) or anticorrelated ($q \approx -1$). It is relevant to note that in both spin and structural glasses an almost constant correlation function such as that of Fig. 3f indicates a dynamical transition where ergodicity is broken[2].

The stability of the RSB process can be addressed acting on the cut-off length $R$, which fixes the maximal spatial scale for intensity correlations. $R$ is related to the number of interacting spatial modes taken into account in the analysis. Increasing $R$ means accounting for modes with a large interaction range and hence exploring the mean-field regime in which all modes are interacting. Figure 4 shows the overlap distribution $P(q)$ for various $R$; we found that the RSB onset weakly depends on the cut-off scale. In the glassy phase, we note a weakening of the peaks in the $P(q)$ for small cut-off distances ($R = 30$ μm and $R = 40$ μm). Moreover, the overlap distribution does not present further significant changes when a cut-off length larger than $R = R_{min} = 100$ μm is considered, which indicates that most of the relevant modes have been included for $R \approx R_{min}$. These facts agree with the behavior of $g(x)$ shown in Fig. 3f; different realizations are not distinguishable at small spatial scales. On the contrary, they are well separated at a scale of the order of 80 μm. This circumstance may indicate a weakening of the glassy behavior when only nearest-neighbor points of field are randomly coupled.

## Discussion

In conclusion, we have reported the observation of replica symmetry breaking for waves propagating in nonlinear disordered media providing a direct measurement of the Parisi overlap distribution function. The glassy phase of light, that emerges as nonlinear interaction overcomes a threshold, is characterized by strong shot-to-shot variations of the speckle-like intensity distribution and the degree of spatial coherence. Surprisingly, these fluctuations are not randomly distributed but can be either completely correlated or anticorrelated, as resulting from separated groups of states in the underlying energy landscape. In agreement with spin-glass theory, the overlap distribution between identical replicas incurs in a nontrivial change that indicates how the same realizations of the dynamics may give rise to statistically different physical observables. This nontrivial change of the overlap distribution is observed although disorder in the system cannot be considered as quenched over all the realizations, a point that remains generally open and whose understanding needs further theoretical efforts[15,16].

These findings are general and do not depend on the specific form and character of the nonlinearity and type of disorder. They can be extended to a large class of optical systems including disordered periodic potentials, such as disordered waveguide arrays[45], speckle patterns from nonlinear scatterers[46–49], multiple pulse filamentation[38,39] and wave propagation in the time domain[50,51]. A particularly promising optical setting may also be found in nonlinear multimode fibers[52,53]; here structural disorder is absent but complex dynamics are known to emerge from the disordered interaction of several excited guided modes. Beyond optics, the universality of the RSB scenario for propagating waves may be found from hydrodynamics to Bose–Einstein condensates[54–56]. Our evidence of a large scale coherence in the presence of disorder at the RSB transition may open a number of further scenarios for testing fundamental physical theories, for example quantum phase transitions in disordered systems.

## Methods

**Experimental setup**. An optical beam at wavelength $\lambda = 532$ nm from a continuous 30 mW Nd:YAG laser source is focused down to a cylindrical Gaussian beam with waist $\omega_0 = 7$ μm along the $y$-direction and quasi-homogeneous (several mm wide) along the $x$-direction. The wave is launched into an optical quality sample of $K_{0.985}Li_{0.015}Ta_{0.63}Nb_{0.37}O_3$ (KLTN)[57], $3.9^{(x)} \times 0.9^{(y)} \times 2.4^{(z)}$ mm size, with a layer of He$^+$ ions implanted beneath its surface. Implantation at 2.3 MeV with a fluence of $0.8 \times 10^{16}$ ions per cm$^{-2}$ yields a partially amorphous layer with refractive index profile as presented in Fig. 1b. This forms a slab waveguide between the surface of the sample with the implanted layer that acts as the cladding[58]. A sketch of the optical system is shown in Fig. 1a. In linear conditions, the line beam experiences no appreciable diffraction along $x$ inside the slab waveguide. The crystal exhibits a ferroelectric phase transition at the Curie temperature $T_C = 285$ K and nonlinear light dynamics is studied at $T = T_C + 2$ K, a condition ensuing the presence of small-scale disorder, giant nonlinearity, optimal transmission and high reproducibility[28]. The input $y$-polarized wave copropagates along the $z$-axis with a uniform background intensity, and the time-dependent photorefractive nonlinearity sets in when an external bias field $E$ is applied along $y$ (Supplementary Note 2). In nonlinear conditions, the involved local variations of the refractive index $\delta n$ are at most of the order of $10^{-3}$ and do not affect the confining slab profile. The spatial intensity distribution is measured at the slab output for different nonlinearity (exposure times) and input power by means of an high-resolution imaging system composed by an objective lens ($NA = 0.5$) and a CCD camera at 15 Hz. The output analyzed area is that corresponding to the imaged slab confining area, ~10 μm and 300 μm wide along $y$ and $x$, respectively.

**Replicated experiments**. Each shot is a single realization of the experiment with fixed experimental parameters; it corresponds to the nonlinear dynamics of the input wave field as observed at the slab waveguide output for increasing non-linearities. To acquire replicas of the optical dynamics the whole experiment is repeated keeping the experimental conditions fixed. Between a shot and the subsequent one the continuous-wave laser is blocked, the sample is heated up to $T_C + 15$ K, it is allowed to relax under mW white-light illumination (microscope illuminator) to clean it from the previously photogenerated charges, and, after 20–30 min, it is slowly brought back to the operating temperature. This procedure is sufficient to erase the photorefractive space-charge pattern written by the previous highly-nonlinear dynamics. No evidence of memory effects is found, so as to have independent realizations. However, the method does not allow us to fix the actual distribution of disorder that arises from sample inhomogeneities due to the proximity of the ferroelectric phase transition (linear index of refraction variations are of the order of $10^{-4}$)[59]. In fact, although these polar-regions may form preferentially in specific spatial points related, for example, either to composition and implantation defects or surface strains, their spatial distribution varies for any realization. Differently, the degree (strength) of disorder can be considered as constant for fixed experimental conditions such as crystal temperature and cooling rate.

**Data availability**. Data supporting the reported results and other findings of this study are available from the corresponding author upon reasonable request.

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

## Acknowledgements

Funding from Sapienza 2016 Research Projects are acknowledged. C.C. acknowledged support from the John Templeton Foundation (58277). A.J.A. acknowledges the support of the Peter Brojde Center for Innovative Engineering.

## Author contributions

D.P., C.C. and E.D. conceived the idea and its experimental realization. D.P., A.T. and F.D.M. carried out the experiments and data analysis. D.P., C.C. and E.D. developed the interpretation of results. D.P. and C.C. developed the theoretical framework. A.J.A. designed and fabricated the slab waveguide sample. All authors discussed the results. D.P., C.C. and E.D. wrote the paper with contributions from all authors.

## Additional information

**Competing interests:** The authors declare no competing financial interests.

