## [Peer Review File · Nature Communications]

Reviewers' comments:

Reviewer #1 (Remarks to the Author):

The article report on the observation of replica symmetry breaking in light propagation through photorefractive thin films near the phase transition temperature when optical non linear propagation is considered. The speckle patterns generated by the light transmitted through the material are examined to search for spatial fluctuation correlations. The autocorrelations overlaps distributions show some change from being essentially uncorrelated ($q=0$ mostly) to being (anti-)correlated ($|q|=1$ mostly).

The authors point that replica symmetry breaking has so far only been observed in Random Lasers emission but seem to forget that disorder needn't be in the structure but can also be in the interactions. An example is provided by Basak, Sci. Rep. 6, 32134 (2016) where this effect is found in a structure where disorder is absent, not a notable characteristic or, at least, not intentionally added.

Although the work is original for the most part, the authors seem to have overlooked, and must include in the bibliography, the three references mentioned here.

Sci. Rep. 5, 16792 (2015)

Sci. Rep. 6, 32134 (2016)

Sci. Rep. 6, 37113 (2016)

The authors chose to study the autocorrelations overlap distributions. They offer this magnitude as a "natural choice" to characterize the replicas. A reader would appreciate a justification of this choice and what other possibilities they ruled out and why. Wouldn't it be more natural to use the bare spatial intensity distributions?

In the supplementary authors claim that "This quantity at the glass transition sharply passes from $Q = 0$ to $Q = 1$. In remarkable agreement with our experimental observations" This statement is far from true. On the one hand the distribution is never purely $q=0$ and on the other the transition is not sharp. It is undeniable that some transition occurs (despite distributions are not showed and only two or three point are plot in Figure 3.)

The work contains a considerable amount of theory however it doesn't provide any phase diagram like Antenucci, Sci. Rep. 5, 16792 (2015) to situate the evolution of the system in the space of phase.

Even, a full classification of the replica breaking is lacking. The authors must declare why they consider that in going from a $q=0$ centered overlaps distribution to a $|q|=1$ centered distribution where (apart from a sign reversal) the distribution is single peaked, it is justified to call that replica breaking. Do they consider it one-step, full, some mix process ...? Isn't it simply a broadened replica symmetric evolution? (see comment about cut off below)

Since the laser used is continuous it is not clear how the replicas are acquired. What defines a "shot"? The methods part is not very clear in that respect. If speckles are acquired sequentially without "erasing" the photorefractive pattern, is there a memory effect? What is the shortest time between two replicas to appear different? Does this much slower process bear any relation with surprising results in non-static systems such as that of Tommasi Sci. Rep. 6, 37113 (2016)?

Is Fig. 1a actually representative of the experiment? What is the area recorded in relation to the focus of the pump? Is there a lateral (x) propagation as well as in depth (z)?

When the authors speaks of non-linearity they impute it to exposure and thus measure it in units of time. This must be better explained. Does exposure not only "randomly" change the distribution of nonlinearity associated with the optical potential in the near phase change material but increase it? What happens for very long times?

Since $P(q)$ is a statistical distribution all figures pertaining to it should present data normalized to unit integral.

This will help see that when the cut off length is reduced the validity of the analysis weakens.

What will become apparent is that overlaps with values around 0.5 are more and more relevant and correlations undefined as a result of part of the modes not being taken into consideration. The authors must determine the minimum cut off length when distribution ceases changing (because no more relevant modes get included).

Reviewer #2 (Remarks to the Author):

In the work authors experimentally observe the replica symmetry breaking in the optical wave propagation in disordered nonlinear photoelectric slab waveguide. They observe signals on a transition to the optical spin-glass case. The obtained results are interesting as for the theory as for the experiments in other areas like BEC with disordered optical potentials. The paper can be recommended to publication by Nature communications.

I have remark about the supplementary material and the main text.

1.The presentation after eq.1 needs in an elaboration. For example authors claim that they consider the limit $|\psi|^2 \gg 1$ in the expression $\Delta n/(1+|\psi|^2)^2$. But it lead to the vanishing nonlinearity?

2.Also should be clarified a connection between the time t and the strength of the nonlinearity in the pictures.

Reviewer #3 (Remarks to the Author):

The authors report on the first experimental evidence of replica symmetry in optical wave propagation. This occurs due to the intricate interaction between disorder and nonlinearity. Under certain conditions, identical "optical glassy states" can have macroscopically different properties.

The manuscript is well written and report on an important results pertaining to replica symmetry-breaking. In this respect, the current manuscript warrants publication. However, I have few comments that the authors need to address:

1. Why there is a need for a nonlinear when it comes to replica symmetry breaking? Can this phenomenon be observed at the linear regime?

2. The effect of disorder on physical system can lead to many interesting dynamics. Why the authors do not see Anderson localization?

3. It was not clear to me if the experimental results are performed for one realization of disorder? Is there an average taken over disorder?

4.How universal the phenomenon of optical replica symmetry breaking is? Could it exists in other optical setting?

=====

Response to Report of Reviewer #1

=====

Report #1: The article report on the observation of replica symmetry breaking in light propagation through photorefractive thin films near the phase transition temperature when optical non linear propagation is considered. The speckle patterns generated by the light transmitted through the material are examined to search for spatial fluctuation correlations. The autocorrelations overlaps distributions show some change from being essentially uncorrelated ($q=0$ mostly) to being (anti-)correlated ($|q|=1$ mostly).

Response: We thank the Reviewer for the detailed report and for interesting comments and requirements that have come to benefit the revised paper.

Report #1: The authors point that replica symmetry breaking has so far only been observed in Random Lasers emission but seem to forget that disorder needn't be in the structure but can also be in the interactions. An example is provided by Basak, Sci. Rep. 6, 32134 (2016) where this effect is find in a structure where disorder is absent, not a notable characteristic or, at least, not intentionally added.

Response: We totally agree with the Reviewer, the fundamental mechanism for replica symmetry breaking is disorder in the interactions. In random lasers structural disorder has a dual role since it provides both the large number of interacting lasing modes and fixes their disordered interactions through their spatial overlaps. However, several interacting modes with disordered couplings can also be obtained with a large leaky cavity, as suggested by the Reviewer and experimentally demonstrated in Sci. Rep. 6, 32134 (2016). Considering the replica symmetry breaking scenario in nonlinear propagation, this fact means that potentially the phenomenon we report can also occur in ordered structures, such as, for example, in nonlinear propagation in multimode fibers, where structural disorder is absent but a complex dynamics can emerge from the strong interplay of several excited guided modes.

We have underlined this point in the introduction of the revised manuscript. (change 1)

Report #1: Although the work is original for the most part, the authors seem to have overlooked, and must include in the bibliography, the three references mentioned here. Sci. Rep. 5, 16792 (2015) Sci. Rep. 6, 32134 (2016) Sci. Rep. 6, 37113 (2016)

Response: Yes, although two of the three suggested references were already present in the original manuscript ([10] in the original main manuscript and [8] in the supplementary), we recognize that they were not sufficiently highlighted with respect to their relevance for the present work.

The suggested references have been added and mentioned in various points of the revisited manuscript. As the number of references in our original submission was limited for editorial reasons that no longer apply, also other relevant references have been added. (change 2)

Report #1: The authors chose to study the autocorrelations overlap distributions. They offer this magnitude as a "natural choice" to characterize the replicas. A reader would appreciate a justification of this choice and what other possibilities they ruled out and why. Wouldn't it be more natural to use the bare spatial intensity distributions?

Response: Yes, this is surely a fundamental point. First of all, it should be stressed that the major hurdle in revealing the breaking of replica symmetry for propagating waves is that the mode eigenvalues cannot be accessed. At variance with random lasers where modes are easily detectable through their emission frequency, here the propagating wave vectors cannot be measured and to gain access to the mode intensities the entire spatial distribution of the field have to be considered. In this framework, a direct quantity to characterize the replicas is, in principle, their bare spatial intensity distributions, as correctly noted by the Reviewer. The reason why this quantity has been ruled out is mainly experimental. Specifically, in evaluating the overlap between two replicas, the intensities should be compared pixel by

pixel, with each pixel corresponding to a specific spatial point (the resolution is approximately 0.3 μm). Since the intensities strongly vary on a small scale of the order of few pixels, the analysis is unstable with respect to the method used to obtain each realization. In fact, for each "shot" the experiment is entirely repeated keeping the experimental conditions fixed (as now detailed in the methods section) and practically it is not possible to maintain the system in the same position on the pixel (sub-micrometric) resolution. Even the thermal expansion of the crystal associated to the heating process would be enough to lose this fine alignment. This fact introduces an arbitrary spatial shift of the output intensity distribution from shot to shot, which acts as an artificial randomization of the spatial coordinate and inevitably alters the overlap evaluation through the local intensity distribution. However, it may be interesting to note that even in this case we have found evidences of a replica symmetry breaking. On the other hand, the intensity autocorrelation is not affected by this experimental limitation as it is an averaged quantity depending only on the relative spatial coordinate. Moreover, mode interaction and locking generally are known to result into typical features of the autocorrelation function (see new Ref. [50] for an optical example in the time domain). For these reasons we consider the spatial autocorrelation as a natural choice to characterize the replicas.

The above points have been discussed in the revised manuscript and supplementary. (change 3)

Report #1: In the supplementary authors claim that "This quantity at the glass transition sharply passes from $Q = 0$ to $Q = 1$. In remarkable agreement with our experimental observations" This statement is far from true. On the one hand the distribution is never purely $q=0$ and on the other the transition is not sharp. It is undeniable that some transition occurs (despite distributions are not showed and only two or three point are plot in Figure 3.)

Response: Yes, we agree, what we wrote was incorrect. The above sentence, as well as the entire paragraph, has been properly revised. (change 4)

Report #1: The work contains a considerable amount of theory however it doesn't provide any phase diagram like Antenucci, Sci. Rep. 5, 16792 (2015) to situate the evolution of the system in the space of phase. Even, a full classification of the replica braking is lacking. The authors must declare why they consider that in going from a $q=0$ centered overlaps distribution to a $|q|=1$ centered distribution where (apart from a sign reversal) the distribution is single peaked, it is justified to call that replica breaking. Do they consider it one-step, full, some mix process ...? Isn't it simply a broadened replica symmetric evolution? (see comment about cut off below)

Response: Yes, we agree, the theoretical details underlying our observations are an extremely interesting argument. However, let us say that the main manuscript is almost entirely experimental; the theoretical framework in the supplementary only serves to provide a background that places the paper in the proper context and assists the not expert reader in avoiding the very technical theoretical literature, whereas a rigorous theoretical understanding of the replica breaking is beyond the scope of the present work and can be found in Refs. [13-15] of the revised manuscript. Being the first observation of replica symmetry breaking for propagating waves, naturally some theoretical points remain open and we hope they will be the subject of active research, as is now happening in random lasers. This said, we have expanded and discussed in more detail the suggested theoretical aspects. The phase diagram reported in Ref. [13], where a spin-glass description for propagating waves is considered, also applies here in the zero-temperature limit. The main qualitative difference with the one reported for open cavities in Ref. [15], Antenucci et al., Sci. Rep. 5, 16792 (2015), is that for nonlinear waves the average nonlinear interaction can be either positive and negative, which corresponds to a defocusing and focusing nonlinearity, respectively. For negative couplings (our case) a transition to a glassy regime is expected increasing the nonlinearity even for small degrees of disorder, whereas in lasing systems without absorbers (positive couplings) generally standard mode locking dominates in these conditions. As for the characterization of the replica breaking, the most reasonable scenario considering our experimental results may be a one-step plus full replica symmetry breaking. In fact, the region at small overlap is not completely depleted as expected for purely one-step replica symmetry breaking, and, at the same time, we are able to identify two distinct behaviors in the autocorrelation functions (Fig. 3(f)). The observation of two well-separated groups of states that are absent for lower nonlinearities allow us to exclude a pure full replica symmetry breaking scenario (which provides a hierarchy of states in phase space), as well as a broadened replica symmetric evolution. This also corroborates with

theoretical results in Ref. [13] where one-step replica symmetry breaking is rigorously demonstrated in the thermodynamic limit for nonlinear optical propagation.

In the revised manuscript and supplementary we have discussed and clarified the above points, further expanding the theoretical section. Specifically, a phase diagram according with Ref. [13] has been added (Supplementary Fig. 1) and the nature of the replica breaking has been discussed. In this case, the attention has been focused on the observed overlap distributions and in connecting our experimental evidence with the detailed classification in Antenucci et al., Sci. Rep. 5, 16792 (2015), where the overlap distribution between mode intensities is considered. (change 4 and change 5)

Report #1: Since the laser used is continuous it is not clear how the replicas are acquired. What defines a "shot"? The methods part is not very clear in that respect. If speckles are acquired sequentially without "erasing" the photorefractive pattern, is there a memory effect? What is the shortest time between two replicas to appear different? Does this much slower process bear any relation with surprising results in non-static systems such as that of Tommasi Sci. Rep. 6, 37113 (2016)? Is Fig. 1a actually representative of the experiment? What is the area recorded in relation to the focus of the pump? Is there a lateral (x) propagation as well as in depth (z)? When the authors speaks of non-linearity they impute it to exposure and thus measure it in units of time. This must be better explained. Does exposure not only "randomly" change the distribution of nonlinearity associated with the optical potential in the near phase change material but increase it? What happens for very long times?

Response: We recognize that the description of the experimental setting, as well as the methods adopted, were not presented clearly enough. In this respect, we have revised the methods section clarifying all the suggested points that may actually confuse the reader. (change 6) An experimental section has been also added in the supplementary. (change 7)

A "shot" is a single realization of the experiment with fixed experimental parameters; it corresponds to the nonlinear dynamics of the input wave field as observed at the slab waveguide output for increasing nonlinearities. Fig. 2(a), for instance, represents a single shot fully-resolved with respect to the nonlinearity. To acquire a replica of this dynamic the whole experiment is repeated keeping fixed the experimental conditions. Between a shot and the subsequent one the principal continuous laser is blocked, the sample is heated up to T_c+15K , it is allowed to relax under mW white light illumination to clean it from the previously photogenerated charges, and, after 20-30 minutes, it is slowly brought back to the operating temperature. This procedure is sufficient to erase the photorefractive space-charge pattern written by the previous highly-nonlinear dynamics. We have no evidence of memory effects; we have uncorrelated, independent experimental realizations. Thus, even two successive replicas may appear different. From this point of view, our results may be closely related with the replica symmetry breaking scenario observed in non-static systems (Tommasi et al. Sci. Rep. 6, 37113 (2016)), where disorder significantly changes among consecutive shots. This is the reason why we never refer to a quenched disorder in discussing our experiments and we show in Fig. 1(c) different realizations of the linear disordered propagation. In fact, in our case, linear disorder arises from sample inhomogeneities due to the proximity of the ferroelectric phase transition and, on average, it is weak (index of refraction variations of the order of 10^{-4}). Although polar-regions may form preferably in specific spatial points related, for example, either to composition and implantation defects or surface strains, their spatial distribution varies for any realization. Therefore, the replica symmetry breaking phenomenology we report has to be considered as averaged over different realizations of the disorder. The key point is that disorder remains fixed in each specific realization of the dynamics.

As for the optical setup, Fig. 1(a) is representative of the experiment but no significant lateral (x) propagation actually occurs on the crystal length. In fact, the focused line beam, obtained by means of a cylindrical lens ($f=150mm$), is approximately $9\mu m$ wide (full width at half maximum) along the y direction and quasi-homogeneous along x (several millimeter wide), so that, in linear conditions, no appreciable diffraction along x occurs inside the slab waveguide. The output area recorded and analyzed is that corresponding to the imaged slab confining area, approximately $10\mu m$ wide along y and $300\mu m$ wide along x.

Concerning the non-instantaneous and cumulative character of the photorefractive nonlinearity, this means that as the light beam impinges on the biased photorefractive crystal it starts to generate an illumination-dependent variation of the index of refraction by means of carrier excitation and their spatial redistribution (the photogenerated carriers are recombined from one shot to the other by means of high-temperature white-light illumination, as discussed above). This process occurs on a slow time scale (seconds for typical intensities) and the magnitude of the nonlinear change generally grows with the exposure time up to a saturation value. In other words, one can think the nonlinear term in Eq.(1) of the supplementary as parametrically dependent on time. This time dependence is well defined once the input beam intensity, applied voltage and temperature have been fixed. Thus, exposure introduces a local nonlinear change of the underlying linear potential that depends on the local intensity and whose average amplitude (nonlinearity strength) increases and saturates on time. For long times the saturation value is reached, that is, the nonlinearity as well as the light distribution reach a stationary state and no further changes take place over time.

Report #1: Since $P(q)$ is a statistical distribution all figures pertaining to it should present data normalized to unit integral. This will help see that when the cut off length is reduced the validity of the analysis weakens. What will become apparent is that overlaps with values around 0.5 are more and more relevant and correlations undefined as a result of part of the modes not being taken into consideration. The authors must determine the minimum cut off length when distribution ceases changing (because no more relevant modes get included).

Response: Yes, we agree, all the statistical distributions have been properly normalized and the behavior of the overlap distribution as a function of the cut-off length is now clearer. The overlap distribution does not present further significant changes when a cut off length larger than $R=100\mu\text{m}$ is considered (the maximum accessible cut-off length in our experiments is $150\mu\text{m}$). (change 8)

=====

Response to Report of Reviewer #2

=====

Report #2: In the work authors experimentally observe the replica symmetry breaking in the optical wave propagation in disordered nonlinear photoelectric slab waveguide. They observe signals on a transition to the optical spin-glass case. The obtained results are interesting as for the theory as for the experiments in other areas like BEC with disordered optical potentials. The paper can be recommended to publication by Nature communications. I have remark about the supplementary material and the main text.

Response: We thank the Referee for the positive report and for the valuable comments that surely benefit the paper.

Report #2: 1.The presentation after eq.1 needs in an elaboration. For example authors claim that they consider the limit $|\psi|^2 \gg 1$ in the expression $\Delta n / (1 + |\psi|^2)^2$. But it lead to the vanishing nonlinearity?

Response: Yes, we agree, the paragraph after Eq.(1) has been revised and clarified. (change 9) Also other parts of the theoretical section of the supplementary have been further elaborated in response to Reviewers comments. (change 4 and change 5)

As for the saturated regime in which experiments are performed, the Reviewer is right, the sentence above is not accurate. The saturated limit means that the input peak intensity is larger than one. On the other hand, the above expression for the nonlinearity refers to the entire field distribution and it depends on the spatial points through the local intensity. Thus, nonlinear propagation is strongly nonlinear where the intensity is low (and rapidly varies) with respect to the background whereas the dynamics is weakly nonlinear in highly-saturated spatial regions.

Report #2: 2.Also should be clarified a connection between the time t and the strength of the nonlinearity in the pictures.

Response: Yes, we understand that this point should be addressed more in detail. In the revised manuscript we have added a supplementary figure (Supplementary Fig. 2) connecting quantitatively the time and the nonlinearity for the cases shown in Fig. 2(a) and Fig. 2(b). (change 10) The behavior of the average index change as a function of time has been evaluated numerically considering the values of the parameters used in the experiments. In fact, a direct measure of the nonlinearity while the nonlinear dynamics is underway is not feasible.

=====

Response to Report of Reviewer #3

=====

Report #3: The authors report on the first experimental evidence of replica symmetry in optical wave propagation. This occurs due to the intricate interaction between disorder and nonlinearity. Under certain conditions, identical "optical glassy states" can have macroscopically different properties. The manuscript is well written and report on an important results pertaining to replica symmetry-breaking. In this respect, the current manuscript warrants publication. However, I have few comments that the authors need to address:

Response: We thank the Referee for the positive report and for the valuable comments that surely benefit the paper.

Report #3: 1. Why there is a need for a nonlinear when it comes to replica symmetry breaking? Can this phenomenon be observed at the linear regime?

Response: This is certainly a key point. To get an understanding, consider that in a generic spin system interaction between each spin is needed to have temperature dynamics; if these interactions are frustrated by disorder, a low temperature glassy phase with replica symmetry breaking can emerge. Thus, disordered interactions between the constituent elements is the basic ingredient for the breaking of replica symmetry and, of course, this interaction can be linear. However, when the picture is transferred to the optical case and spins are replaced by optical modes, we have to take into account that optical modes do not linearly interact at all if they are orthogonal (as in a closed electromagnetic cavity, where they oscillates independently) and, in more general cases, only weakly interact if the linear perturbation is weak. In these cases, strong interaction between optical modes is always mediated by a nonlinear polarization. In other words, although the spin-glass Hamiltonian (Eq. (6)) would have a glass transition also for a large value of the average linear coupling compared to the nonlinear one, this condition cannot actually occur for optical waves in weak optical potentials and the phenomenon can be observed only in a nonlinear regime.

We have added a statement clarifying this fundamental point. (change 11)

Report #3: 2. The effect of disorder on physical system can lead to many interesting dynamics. Why the authors do not see Anderson localization?

Response: Anderson localization is a linear phenomenon and generally occurs when disorder is strong enough to significantly alter the photon mean free path. In our case, disorder from ferroelectric inhomogeneities in the medium is weak and linear modes are mainly delocalized. This is evident from the output intensity distribution in linear conditions (Fig. 1(c) for example), where we have no traces of either strong or transverse localization. The question is however interesting as one cannot exclude that localization effects resulting from disorder and noise may play a role during propagation in determining that spatial features specifying each realization of the nonlinear dynamics (see M.F.Saleh et al., Opt. Expr. 25, 5457 (2017) for a theoretical and numerical study in the time domain).

Report #3: 3. It was not clear to me if the experimental results are performed for one realization of disorder? Is there an average taken over disorder?

Response: This is an important aspect of our experimental observation of replica symmetry breaking and thus we have made it more clear both in the main manuscript and method section. (change 6)

The method adopted to obtain replicas of the nonlinear propagation dynamics, which consists in entirely repeating uncorrelated experiments as discussed in more detail in the revised manuscript, does not allow us to fix the same distribution of disorder for all the realizations. This is the reason why we never refer to a quenched disorder in discussing our experiments and we show in Fig.1(c) different realizations of the linear disordered propagation. In fact, in our case, weak disorder arises from sample inhomogeneities due to the proximity of the ferroelectric phase transition. Although these polar-regions may form preferably in specific spatial points related, for example, either to composition and implantation defects or surface strains, their spatial distribution varies for any realization. Therefore, even if disorder is not completely random and its strength is fixed for fixed experimental conditions such as temperature and cooling, the replica symmetry breaking phenomenology we report has to be considered as averaged over different realizations of the disorder. We remark that disorder does not change during the single realization of the dynamics. In this respect our results may be closely related with the replica symmetry breaking scenario observed in non-static systems (Tommasi et al. Sci. Rep. 6, 37113 (2016), where disorder significantly changes among consecutive shots).

Report #3: 4.How universal the phenomenon of optical replica symmetry breaking is? Could it exist in other optical settings?

Response: We really thought about it and we believe the phenomenon can be quite universal also beyond optics, as we have remarked in the main manuscript. In optics, we believe that replica symmetry breaking for nonlinear disordered waves can be observed in several different settings. Specifically, the scenario can be extended to generic disordered nonlinear propagation independently from the local/nonlocal and focusing/defocusing character of the nonlinearity; from disordered liquid crystals and systems composed by nonlinear scatterers to disordered waveguide arrays. We are particularly convinced that similar results can also be found in multiple filamentation for high-power pulses propagating in gases. Moreover, in our opinion the phenomenon can also be transferred to the time domain, for example considering disordered nonlinear fibers and, more interestingly, nonlinear multimode fibers. In fact, this last topic is currently attracting a lot of attention by virtue of the complex dynamics emerging from the nonlinear interaction of different guided modes.

In the conclusions of the revised manuscript these perspectives have been further underlined. (change 12)

REVIEWERS' COMMENTS:

Reviewer #1 (Remarks to the Author):

I appreciate the big effort put into the responses and feel satisfied with the answers. Some of them are in fact very elaborated and instructive.

Reviewer #2 (Remarks to the Author):

Authors take into account my remarks properly. The paper can be recommended to the publication by Nature Communications

Reviewer #3 (Remarks to the Author):

The authors revised the manuscript correspondingly and I now recommend publication.